A UK general practice population cohort study investigating the association between lipid lowering drugs and 30-day mortality following medically attended acute respiratory illness

Joshi Roshni 1 2
Venkatesan Sudhir 1 sudhir.venkatesan@nottingham.ac.uk
Myles Puja R. 1
1 Division of Epidemiology and Public Health, University of Nottingham , Nottingham , United Kingdom
2 Quality Standards and Indicators Programme, National Institute for Health and Care Excellence , Manchester , United Kingdom
Kuhle Stefan
Electronic publication date: 2016 Apr 18
Publication date: 2016
Volume: 4
Electronic Location ID: e1902
Received 2015 Nov 20; Accepted 2016 Mar 15
Copyright: ©2016 Joshi et al.
Copyright year: 2016
Copyright holder: Joshi et al.
License: This is an open access article distributed under the terms of the Creative Commons Attribution License, which permits unrestricted use, distribution, reproduction and adaptation in any medium and for any purpose provided that it is properly attributed. For attribution, the original author(s), title, publication source (PeerJ) and either DOI or URL of the article must be cited.
License URL: https://creativecommons.org/licenses/by/4.0/

Keywords: Fibrates, Lipid lowering drugs, statins, mortality, MAARI, Acute respiratory illness, Cohort study, CPRD

Funding: F Hoffman-La Roche educational grant for research in pandemic influenza PRM is the recipient of the unrestricted educational grant for research in the area of pandemic influenza from F Hoffman-La Roche. The authors received no funding for this work.

==============================
Background. Cholesterol lowering drugs HMG-CoA reductase inhibitors (statins) and PPARα activators (fibrates) have been shown to reduce host inflammation via non-disease specific immunomodulatory mechanisms. Recent studies suggest that commonly prescribed drugs in general practice, statins and fibrates, may be beneficial in influenza-like illness related mortality. This retrospective cohort study examines the association between two lipid lowering drugs, statins and fibrates, and all-cause 30-day mortality following a medically attended acute respiratory illness (MAARI).

Methods. Primary care patient data were retrospectively extracted from the UK Clinical Practice Research Datalink (CPRD) database. The sample comprised 201,179 adults aged 30 years or older experiencing a MAARI episode. Patient exposure to statins or fibrates was coded as separate dichotomous variables and deemed current if the most recent GP prescription was issued in the 30 days prior to MAARI diagnosis. Multivariable logistic regression and Cox regression were used for analyses. Adjustment was carried out for chronic lung disease, heart failure, metformin and glitazones, comorbidity burden, socio-demographic and lifestyle variables such as smoking status and body mass index (BMI). Statistical interaction tests were carried out to check for effect modification by gender, body mass index, smoking status and comorbidity.

Results. A total of 1,096 (5%) patients died within the 30-day follow up period. Of this group, 213 (19.4%) were statin users and 4 (0.4%) were fibrate users. After adjustment, a significant 35% reduction in odds [adj OR; 0.65 (95% CI [0.52–0.80])] and a 33% reduction in the hazard [adj HR: 0.67 (95% CI [0.55–0.83])] of all-cause 30-day mortality following MAARI was observed in statin users. A significant effect modification by comorbidity burden was observed for the association between statin use and MAARI-related mortality. Fibrate use was associated with a non-significant reduction in 30-day MAARI-related mortality.

Conclusion. This study suggests that statin use may be associated with a reduction in 30-day mortality following acute respiratory illness that is severe enough to merit medical consultation. Findings from this study support and strengthen similar observational research while providing a strong rationale for a randomised controlled trial investigating the potential role of statins in acute respiratory infections.

Introduction

Globally, an estimated 3.9 million people are killed annually due to acute respiratory infections (ARI). Moreover, specific respiratory infections such as influenza and respiratory syncytial virus (RSV) are major contributors to the mortality and burden from ARI (Legand, Briand & Shindo, 2014). Statins are competitive inhibitors of 3-hydroxy-3-methylglutaryl coenzyme A (HMG-CoA) reductase and are used as lipid lowering drugs to reduce plasma cholesterol (Kwak et al., 2000) and a reduction in chronic inflammation has been reported following their administration in hypercholesterolaemic and normocholesterolaemic individuals (Vaughan, Murphy & Buckley, 1996). In vivo studies demonstrating the ability of statins to repress MHC-II expression, inhibit T cell activation, and limit the release of pro-inflammatory cytokines further suggest that statins have immunosuppressive and immunomodulatory applications (Mach, 2002; Raggatt & Partridge, 2002). Fibrates (bezafibrate, ciprofibrate, fenofibrate and gemfibrozil) are prescribed as first-line therapy in patients with severe hypertriglyceridaemia (Miller & Spence, 1998). ARIs trigger a host inflammatory immune response and can cause excessive secretion of pro-inflammatory cytokines resulting in a cytokine storm, which can increase the risk of complications and mortality. Statins and fibrates could potentially interfere with molecular pathways in influenza infection, reducing secretion of pro-inflammatory cytokines, reducing inflammation, limiting the onset of a cytokine storm and thus potentially reducing morbidity and mortality (Fedson, 2006; Jain & Ridker, 2005). These properties of statins and fibrates could be advantageous in the clinical treatment and management of ARIs especially as they have been used in clinical practice for years and have well established safety profiles. Therefore the aim of this study was to investigate the association between two lipid lowering drugs, statins and fibrates, and all-cause 30-day mortality following a medically attended acute respiratory illness (MAARI).

Methods

Data source

This study uses data obtained from the Clinical Practice Research Datalink (CPRD), a large population based observational and interventional research service providing anonymised UK primary care patient data. General practices that choose to participate in CPRD are required to record all instances of clinical diagnoses, morbidity events, abnormal test results and therapeutic information including prescription information, dosages and methods of administration. Furthermore, additional information is also recorded, such as weight, height, blood pressure measurements and lifestyle factors (Herrett et al., 2010), making it the ideal data source for this study. Ethical approval for research involving CPRD data for this study was obtained from the CPRD Independent Scientific Advisory Committee (ISAC) (ISAC Protocol Number: Protocol 11_14R).

Figure 1 Diagrammatic representation of retrospective cohort study design.

Study design and population

The CPRD data represent a primary care patient cohort and this study was conceptualised as a retrospective cohort study, in which nested case-control and survival analyses were conducted to investigate the research question (Fig. 1). The source population for the present study consisted of all patients registered with general practices contributing to CPRD.

Following a preliminary exploration of prescribing patterns for statins and fibrates, the study period was defined as 1st January 2008 to 31st December 2013. All participants aged 30 years and older with a record of MAARI (Appendix S1) within the study period were included in the present study sample and within this group, patients classified as current statin users based on prescription records were identified. It is assumed that all patients in the study sample that were treated with statins, were clinically indicated for satin prescription (most likely for primary or secondary cardiovascular disease prevention) and that statin prescription implies the use of statins. Those without statin prescription were assumed to be non-users.

The most recent MAARI diagnosis date was used for each patient and even though MAARI could recur during the study period, each patient was only counted once in the study. The start of the 30-day follow up began from the MAARI index date (date of MAARI event as recorded by the physician).

Data variables

The individual’s exposure to either statins or fibrates in the 30 days prior to the MAARI episode index date was coded as separate binary variables (yes/no). The outcome of interest was all-cause mortality occurring in the 30 days following the MAARI index date. From a list of covariates related to both the exposure and outcome of interest, we evaluated the following comorbidities as potential confounders: myocardial infarction, heart failure, peripheral vascular disease, chronic lung disease and hypertension (all coded as dichotomous variables). In addition we adjusted for total comorbidity burden using a weighted Charlson’s comorbidity index (CCI) (Schneeweiss & Maclure, 2000). The CCI scores thus derived were further categorised into 4 levels (0, 1–2, 3–5, >5) for inclusion in the multivariable analysis as a categorical variable. We adjusted for the following drug covariates: glitazones, metformin, beta blockers, ACE inhibitors and angiotensin receptor blockers (ARBs). Finally, we included age, sex, body mass index (BMI) and smoking status as socio-demographic and lifestyle variables, while the presence of an HbA1c measurement was used as a proxy measure of healthcare seeking behaviour (Appendix S1 includes detailed variable definitions). For chronic conditions, we considered a diagnosis of that particular condition at any point prior to the MAARI episode index date. For drug covariates, only current exposures were considered. ‘Current’ was defined as the most recent prescription in the 30 days prior to the MAARI index date.

These covariates were selected for adjustment based on a combination of what other researchers had suggested/used, consultation with clinicians and clinical indications as per the British National Formulary and NICE guidelines (British National Formulary, 2014; NICE clinical guideline 181, 2014).

Given that statin prescription is a choice, apart from the medical indications for statin therapy, certain behavioural factors may be related to statin use. Factors related to statin use such as underlying cardiovascular comorbidity are easier to record, measure and adjust for. However, behavioural factors and lifestyle preferences related to statin use are more difficult to measure accurately. We adjust for the covariates that we have been able to measure (described above). We discuss our findings in the context of those variables that we may not have been able to measure and adjust for.

For variables with <5% missing data, a complete case analysis approach was adopted; for variables with >5% of missing data, a dummy variable was created to represent missing data.

Analysis

Descriptive analysis to summarise data characteristics, identify potential anomalies and quantify missing data was conducted. To assess the average treatment effect across patients treated with statins and fibrates, multivariable logistic regression and Cox regression models were constructed to investigate the association between statins and fibrates and 30-day mortality following MAARI. Proportional hazard assumptions were checked using log–log plots and the Schoenfeld global test. Collinearity was assessed using the variance inflation factor. An a priori decision was taken to include age, sex and current metformin and glitazone use (based on previously reported immunomodulatory activity and likelihood of co-prescription for diabetes mellitus (Fedson, 2009)) in all multivariable models regardless of statistical significance. The models were constructed as follows: Model A included a priori variables, all drug covariates, all comorbidity variables, CCI scores and socio-demographic and lifestyle variables. Model B included a priori variables, and variables independently associated (statistically significant at P ≤ 0.05) with both outcome and exposure. Model C included a priori variables and variables that were both significantly (P ≤ 0.05) associated with 30-day mortality and changed the crude measure of effect by ≥10%. Results are presented as odds ratios (OR), hazard ratios (HR) and 95% confidence intervals (CI). Effect modification was assessed using the likelihood ratio test and Model C was re-run stratified by significant interaction terms. Additionally, we performed a sensitivity analysis where we adjusted for the number of GP visits (included as a covariate) in each of the three models—A, B and C.

All analyses were carried out in Stata 13 (StataCorp. 2009. Stata Statistical Software: Release 11. College Station, TX, USA: StataCorp LP).

Results

The final analysis sample after excluding patients aged 30 years or younger was 201,179 who had a MAARI episode from 2008 to 2013. Of the study population 200,083 (95%) survived at the end of 30-day follow up, of which 40.8% were males and had a median age of 52. Of the surviving group, 27,095 were statin users and 611 were currently using fibrates. Crude analysis showed a significant increased association between statin exposure and 30-day mortality [crude OR: 1.55 (95% CI [1.34–1.81])] (Table 1). All disease variables were significantly associated with 30-day mortality as were all socio-demographic and lifestyle variables.

Table 1 Comparison of patient characteristics among non-statin users and current statin users.

Patient characteristic	Non-statin users (n = 174,084)	Current statin users (n = 27,095)	Unadjusted ORa(95% confidence interval)	P valueb	
Median age (IQR)c	49 (39–63)	69 (60–77)	1.06 (1.06–1.07)	<0.001	
Sex					
Males	68,108 (39.1%)	13,962 (51.5%)	1		
Females	105,976 (60.9 %)	13,133 (48.6%)	0.60 (0.59–0.62)	<0.001	
Hypertension	29,086 (16.2%)	13,377 (49.4%)	4.86 (4.73–4.99)	<0.001	
Myocardial infarction	3,134 (1.9%)	3,774 (13.9%)	8.34 (7.94–8.75)	<0.001	
Heart failure	1,410 (0.8%)	958 (3.5%)	4.49 (4.13–4.88)	<0.001	
Peripheral vascular disease	1,076 (0.6%)	991 (3.7%)	6.10 (5.59–6.66)	<0.001	
Chronic lung disease	31,689 (18.2%)	6,041 (22.3%)	1.29 (1.25–1.33)	<0.001	
Diabetes	10,162 (5.8%)	8,901 (32.9%)	7.89 (7.64–8.15)	<0.001	
Charlson’s comorbidity score					
0	135,609 (77.9%)	10,315 (38.1%)	1		
1–2	29,257 (16.8%)	10,707 (39.5%)	4.81 (4.67–4.96)		
3–5	7,021 (4.0%)	4,476 (16.5%)	8.39 (8.03–8.75)		
>5	2,197 (1.3%)	1,597 (5.9%)	9.56 (8.93–10.22)	<0.001	
Fibrates	363 (0.2%)	248 (0.9%)	4.42 (3.76–5.20)	<0.001	
Glitazones	233 (0.1%)	690 (2.6%)	19.50 (16.80–22.63)	<0.001	
Metformin	2,009 (1.2%)	4,608 (17.0%)	17.55 (16.63–18.53)	<0.001	
Beta blockers	3,452 (2.0%)	4,310 (15.9%)	9.35 (8.92–9.80)	<0.001	
ARB	4,056 (2.3%)	4,089 (15.1%)	7.45 (7.12–7.80)	<0.001	
Smoking status					
Never-smoker	21,812 (18.7%)	2,627 (14.7%)	1		
Ex-smoker	66,505 (57.4%)	8,475 (47.4%)	1.06 (1.01–1.12)		
Current-smoker	27,654 (23.9)	6,765 (37.9%)	2.03 (1.94–2.13)	<0.001f	
BMId					
Underweight	2,433 (2.3%)	251 (1.5%)	1		
Normal weight	38,461 (36.5%)	3,817 (22.2%)	0.97 (0.84–1.10)		
Overweight	37,522 (35.6%)	6,815 (39.6%)	1.77 (1.55–2.02)		
Obese	26,983 (25.6%)	6,295 (36.7%)	2.27 (1.99–2.59)	<0.001f	
			Mean difference (95% CI)	p-value	
Mean number of GP consultations(SD)e	213.06 (165.25)	376.82 (229.27)	163.76 (−166.01–161.52)	<0.001	
Notes.

a Odds ratio.

b Wald’s p values.

c Interquartile range.

d Body mass index.

e Standard Deviation.

f Wald’s p value for trend.

Significant p values shown in bold.

All three multivariable logistic regression models yielded statistically significant point estimates ranging from 0.63 to 0.67 as shown in Table 2. There was no effect modification of the association between statins and 30-day mortality by either gender or BMI. However, a significant interaction was found for CCI scores and therefore, in line with the analysis strategy, Model C was re-run stratified by CCI score categories. The results of stratification showed point estimates ranged from 0.48 to 0.63 but with overlapping 95% confidence intervals (Table 3).

Table 2 The association between statins and 30-day mortality following MAARI.

	ORa	95% CIb	P-Value	
Crude	1.55	1.34–1.81	<0.001	
Model Ac	0.67	0.54–0.83	<0.001	
Model Bd	0.63	0.51–0.78	<0.001	
Model Ce	0.65	0.52–0.80	<0.001	
Notes.

a Odds Ratio

b Confidence Interval

c Adjusted for a priori confounders, all comorbidity variables, all drug covariate variables, all potential confounding variables

d Adjusted for a priori confounders, variables significantly associated with both outcome and exposure (≤0.05)

e Adjusted for a priori confounders, variables significantly associated with both outcome and exposure and altering the crude OR by ≥10%

Variables included in models A, B and C detailed in Appendix S2

Significant P- values shown in bold

Table 3 The association between statins and 30-day mortality following MAARI: stratification by Charlson’s Comorbidity Index categories.

	Adjusted ORa	95% CIb	P value	
Comorbidity index score 0	0.63	0.45–0.89	0.008c	
Comorbidity index score 1–2	0.57	0.43–0.79	0.001	
Comorbidity index score 3–5	0.48	0.43–0.79	<0.001	
Comorbidity index score >5	0.73	0.48–1.09	0.126	
Notes.

a Odds Ratio

b Confidence Interval

c LRT p value

Significant p values shown in bold

The proportional hazards assumption was fulfilled as determined using log–log plots and the Schoenfeld global test. Crude analysis found an increase in the hazard for 30-day mortality of 58% in the statin users group [crude HR: 1.58 (95% CI [1.36–1.83])]. Multivariable Cox proportional hazard regression analyses were conducted using the previously outlined multivariable model building strategy and yielded significant point estimates ranging from 0.66 to 0.70 (Table 4).

Table 4 Hazard ratios (95% CI) representing the association between statin exposure and 30-day mortality following MAARI.

	HRa	95% CIb	P Value	
Crude	1.58	1.36–1.83	<0.001	
Model Ac	0.70	0.57–0.83	0.001	
Model Bd	0.66	0.53–0.81	0.001	
Model Ce	0.67	0.55–0.83	<0.001	
Notes.

a Hazard Ratio

b Confidence Interval

c Adjusted for a priori confounders, all comorbidity variables, all drug covariate variables, all potential confounding variables

d Adjusted for a priori confounders, variables significantly associated with both outcome and exposure (≤0.05)

e Adjustedfor a priori confounders, variables significantly associated with both outcome and exposure and altering the crude HR by ≥10%

Variables included in models A, B and C detailed in Appendix S2

Significant p values shown in bold

Sensitivity analysis

Statin users in our study sample were more likely (p < 0.001) to consult GPs than non-users (Table 1). We performed a sensitivity analysis where we adjusted for number of GP consultations in our multivariable models. After adjustment of number of GP visits, we obtained very similar estimates to our primary results: Model A, OR (95% CI): 0.67 (0.54–0.84); Model B: 0.63 (0.51–0.78); Model C: 0.65 (0.52–0.80). All three estimates were statistically significant (p < 0.001).

Discussion

Summary of main findings

The findings of this study suggest that the use of statins is associated with decreased mortality in the 30 days following a MAARI. Following adjustments for myocardial infarction, hypertension, heart failure, diabetes, CCI and BMI (fully adjusted model C), current statin use was protective and decreased the odds of MAARI-related mortality by 30%. Cox regression analysis showed a similar decrease in 30-day mortality hazard among statin users of 35%. We found a significant interaction for the effect of statins with comorbidity as measured using the CCI and while stratification showed significant protective effects of statins in the lower comorbidity categories, the 95% confidence intervals were overlapping suggesting no clinically meaningful effect modifications. Fibrate use was found to decrease MAARI-related mortality, but the results were non-significant.

Strengths and limitations

The validity of this study is enhanced by the quality, comprehensiveness and representativeness of the data recorded within CPRD (Herrett et al., 2010). Additionally, the large sample exceeds a priori sample size estimates (Appendix S3), therefore increasing statistical power and reducing type II errors. Moreover, utilisation of contemporary data makes the findings potentially applicable to current prescribing patterns. The retrospective cohort design of this study eliminates temporal bias and there is no risk of recall bias as data are prospectively added to CPRD and there is no risk of reverse causation as the outcome of death is final. It is however, possible that miscoded entries due to user error may cause systematic errors and could result in non-differential misclassification bias, pushing findings towards the null hypothesis.

All General Practitioner (GP) issued prescriptions are recorded within the database thus minimising misclassification bias. While prescriptions issued in secondary care facilities are not accounted for in CPRD, long-term medication for chronic conditions like statins, fibrates, glitazones and metformin are mostly prescribed in primary care. Moreover, prescriptions are a proxy measure of drug use and assume patient compliance. Lack of compliance would lead to misclassification of exposure status and lead to an over-estimation of the association observed and push findings towards the null hypothesis. The present study assumes patient compliance to prescriptions based on indirect evidence that most patients do take prescribed drugs especially long-term medication (Jick et al., 2003). Moreover, this study assumes that those without a prescription for statins are non-users, however it is possible that individuals who have a clinical indication for receiving stations but refuse treatment have been omitted from the study sample and could therefore overestimate the measure of effect. Furthermore, this study assumes that although there may be differences in patients who were prescribed statins and those who were not, i.e., confounding by indication leading to an overestimation of the observed effect, these differences would be accounted for in the multivariable regression models.

Selection bias is minimised due to the method of data collection in CPRD, however a possible limitation of CPRD data analysis is that MAARI patients presenting directly to secondary institutions (and presumably, more severe cases) may not be included, and therefore, these findings may only be applicable to those with less severe MAARI that can be managed in primary care.

Finally, it is not possible to prove a causal relationship between statin use and MAARI related mortality based on an observational study; to strengthen the causal inference we adjusted for a variety of drug, disease, socio-demographic and lifestyle covariates. However, an important limitation with observational studies is that of residual confounding and omitted variable bias. It is possible that, despite our attempts to adjust for relevant confounders, incorrect measurement of a particular independent variable or omission of an unknown confounding variable could affect our estimates. We attempted to overcome confounding by indication by estimating propensity scores. However, our propensity scores could be limited by potentially omitted variables.

We earlier stated that there may be behavioural factors related to statin use that we may not have been able to measure and subsequently adjust for in our analysis. There is evidence to suggest that statin users are more likely to adopt a healthcare seeking lifestyle resulting in them being healthier than non-users (Brookhart et al., 2007). This ‘healthy user effect’ (i.e., statin users are healthier than non-users) has been proposed as one likely explanation for previously observed statin-related benefits in infection. However in our study sample, statin users were significantly more likely to have a history of hypertension, myocardial infarction, heart failure, peripheral vascular disease, chronic lung disease and higher number of comorbidities; statin users were also more likely to be current smokers and obese (Appendix S4). These factors explain why the crude effect estimates showed a higher likelihood of MAARI-related mortality in statin users. In line with previous evidence (Brookhart et al., 2007), statin users in our study sample were more likely to visit their physicians. However, when we adjusted for the number of GP visits, our estimates did not change substantially. It is also important to note that the increased number of GP visits among statin users may reflect genuine sicker patients with increased healthcare needs rather than being indicative of a healthier lifestyle and it is unlikely that we have been able to fully account for the healthy user effect. However, this study doesn’t account for people who may have had elevated cholesterol, but were contraindicated for statin therapy because they were likely to experience side effects, not likely to take or adhere to prescriptions or were too frail. Patient frailty, although difficult to measure in study, has not been accounted for, contributing to residual confounding and should be taken into consideration when interpreting the findings from this study. Finally, we have not performed instrumental variable analysis, a method described in the literature (Polgreen et al., 2015) to account for non-random treatment assignment in observational studies investigating treatment effects, in this current study; this is a limitation of this study.

Comparison with previous work

Statins and mortality

Existing research concurs with the protective association of statin use and MAARI-related mortality results from this study (Frost et al., 2007; Myles et al., 2014). Similarly, data from the Centre for Disease Control and Prevention (CDC) influenza hospitalisation surveillance system (Vandermeer et al., 2012), showed a similar yet more conservative protective effect was reported, [adj. OR 0.59 (95% CI [0.38–0.92])] and could be due to Vandermeer et al. adjusting for age and comorbidity variables in-line with those included in the present study as well as, influenza vaccination status and race; additional variables unaccounted for in this study. Moreover, Vandermeer and colleagues considered patients requiring hospitalisation for influenza hence were more seriously ill and therefore the capacity to benefit is likely to have been greater. Similarly, Myles et al. reported a significant reduction in pneumonia-related mortality among current statin users using data extracted from The Health Improvement Network (THIN), a UK primary healthcare database similar to CPRD and could not find evidence for the ‘healthy user effect.’ Finally a Canadian retrospective cohort study (Kwong, Li & Redelmeier, 2009) linking multiple administrative health-care databases over a 10 year period found borderline protective effects of statin exposure in relation to 30-day mortality following influenza diagnosis [crude OR: 0.92 (95% CI [0.89–0.95])] and a larger protective effect following pneumonia diagnosis [crude: OR 0.84 (95% CI [0.77–0.91])]; however, both estimates shifted towards the null following adjustment. This could be due to misclassification of exposure status as statin use was not captured during hospitalisation of patients.

One randomised controlled trial performed in intensive care units in France, investigating the effect of simvastatin treatment on mortality in patients with ventilator-associated pneumonia reported no significant difference in 28-day mortality between the statin and the placebo groups [Hazard Ratio (HR): 1.45 (95% CI [0.83–2.51])] (Papazian et al., 2013). Another trial performed in hospitalised patients with community-acquired pneumonia showed no significant difference in time from hospital admission to clinical stability between patients treated with simvastatin and placebo (median: 3 days (IQR: 2–5) vs 3 days (IQR: 2–5); p-value: 0.47)) (Viasus et al., 2015). The 2012 JUPITER trial studied the effect of rosuvastatin treatment on incident pneumonia in healthy patients reported a modest benefit of statin treatment on the incidence of pneumonia (Novack et al., 2012).

While previous observational studies have reported a protective effect of statins, evidence from randomised controlled trials show little of no effect of statins on pneumonia. One explanation for this difference in estimates could be that the observational studies described above were carried out mostly using primary care datasets, whereas the trials have been carried out in intensive care units and hospitalised patients where patients were demonstrably more ill. Polgreen et al., (2015) in their cohort study investigating the effect of statins on pneumonia, have tried to minimise limitations of an observational study design by performing instrumental variable analysis to account for non-random assignment of statin treatment. They reported that while statins were associated with a reduction in pneumonia incidence in their initial analysis, this protective effect of statins was not seen after accounting for non-random statin assignment indicating that the protective effect seen initially was most likely due to healthy user effect (Polgreen et al., 2015).

The findings in this study suggest that statins may confer mortality-reduction benefits in patients with MAARI. However, it should be noted that observational studies alone cannot prove a conclusive causal relationship between statin exposure and decreased 30-day mortality following MAARI and associations observed may be attributable to residual confounding; therefore, the findings of this study should be considered in the context of other studies across different populations and animal studies when considering the clinical implications. Nonetheless, the observed mortality reduction among statin users are biologically plausible as their immunomodulatory action has been demonstrated in animal and laboratory studies (Raggatt & Partridge, 2002; Liao & Laufs, 2005; Farmer, 2000; Davignon, 2004) and therefore, they could benefit MAARI patients by mediating their immune response. However, prior to widespread use in this context, especially among patients in whom statins would not otherwise be clinically indicated, randomised control trials are required to confirm these potential benefits. It would also be valuable to investigate whether the mortality reduction benefits vary for the different types of statins, the duration of statin exposure and, explicit categorical definitions of statin dosage.

Supplemental Information

Appendix S1 The association between lipid lowering drugs (statins and fibrates) and 30-day mortality following medically attended acute respiratory illness (MAARI) dataset description (n = 201,179)

Click here for additional data file.

Appendix S2 Statin esposure models- Variables included in logistic regression and cox proportional hazards regression analysis

Click here for additional data file.

Appendix S3 Sample size calculations for exposed and unexposed patients to statins and fibrates

Click here for additional data file.

Appendix S4 Comparison of patient characteristics among survivors and non-survivors

Click here for additional data file.

Additional Information and Declarations

Competing Interests

Author Contributions

Human Ethics

Data Availability

SV was involved in this research on pandemic influenza with PRM.

Roshni Joshi performed the experiments, analyzed the data, contributed reagents/materials/analysis tools, wrote the paper, prepared figures and/or tables, reviewed drafts of the paper.

Sudhir Venkatesan analyzed the data, contributed reagents/materials/analysis tools, wrote the paper, reviewed drafts of the paper.

Puja R. Myles conceived and designed the experiments, analyzed the data, contributed reagents/materials/analysis tools, wrote the paper, reviewed drafts of the paper.

The following information was supplied relating to ethical approvals (i.e., approving body and any reference numbers):

Ethical approval for research involving CPRD data for this study was obtained from the CPRD Independent Scientific Advisory Committee (ISAC) (ISAC Protocol Number: Protocol 11_14R).

The following information was supplied regarding data availability:

The research in this article did not generate any raw data. We used existing data from the Clinical Practice Research Datalink on application. Reviewers and interested researchers may apply directly to kc@cprd.com (ISAC protocol number:11_14R) for a copy of these data. The coding lists generated by the authors of the submitted manuscript have been provided in Appendix S1.

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
