# Peer review of "A UK general practice population cohort study investigating the association between lipid lowering drugs and 30-day mortality following medically attended acute respiratory illness"

_PeerJ, doi:10.7717/peerj.1902_

## Round 0.1 · original submission · Major Revisions

Please ensure you address all of the reviewers' comments in the revised article, and include a point by point description of the changes that have been made.

Reviewer 1 ·

Basic reporting

Joshi et al conducted a retrospective cohort study to investigate the effect of lipid lowering drugs on mortality following medically attended respiratory illness with a large registry based database. They found the patients with prior lipid lowering drugs use had significantly lower mortality in 30 days follow up. The method is straightforward and clear. The conclusion is as expectation and compatible with previous observation and literature. Several minor point should be considered.
1. In table 1: What does HbA1C refer to? did it present the measurement or value ?
2. In the adjustment model 3. Since the Charlson’s comorbidity index includes some adjusted co-morbidity such as myocardial infraction, heart failure, etc., the adjustment model may be over-adjusted.
3. Although the authors adjusted the co-morbidities in the statistical models, the "adjustment" may be incomplete. For example, the patient with "good" or "poor" control of DM may affect the results significantly. This issue should be addressed in the discussion.
4. Is the data regarding socioeconomic status available? This information may be important in regression model.
5. In Figure 1. Why are the lengths of two orange lines not equal?

Experimental design

No comments.

Validity of the findings

No comments.

Additional comments

No comments.

Reviewer 2 ·

Basic reporting

This submission adheres to your basic reporting guidelines, except their literature review is incomplete. Specifically, they need to add references that show that "healthy-user bias" exists, and they need to give the results of related randomized controlled trials that have been done.

The article is generally well written, but in the comments to the author I pointed out some unclear sentences that need to be changed.

Experimental design

This is an interesting topic. The research methods here are described well, and as reported, are done appropriately. The research was ethical and approved by the appropriate authorities. The only problem that these researchers have is possible omitted variable bias. For example, they cannot tell how frail these patients are. This is a problem with most observational studies, however, and they simply need to address it more fully.

Validity of the findings

Due to the possible omitted variable bias mentioned above, the conclusions should be a little less firm. As mentioned earlier, there are some gaps in the literature review. Including a wider range of previous findings would make the results a little less firm, but it would make a better manuscript.

Additional comments

This manuscript describes an analysis of outpatient, administrative data and the authors show that controlling for many clinical variables, statins protect patients from mortality following respiratory illness. This paper is generally well written and the analysis described of this observational record is sound. However, there are some gaps in the literature review, and I believe that the conclusions are less justifiable than presented without a more detailed limitations section.
Major problems:
1. A major limitation with all observational studies is omitted variable bias. The authors have a large sample, and many important variables, but this may not be enough to control for important patient differences. The authors dismiss “healthy-user bias” as a reason for the observed protective effects of statins. They state that the statin users in their cohort are demonstrably less healthy than the non-statin users. However, this is because they have a general cohort of people. Statins are not prescribed to people without elevated cholesterol but they are NOT necessarily prescribed to all people with elevated cholesterol. In fact, among patient with elevated cholesterol, only patients expected to benefit from cholesterol-lowering therapy should be prescribed therapy. Thus, patients who might benefit from cholesterol-lowering therapy but are too frail, not likely to take the drug or more likely to experience side effects are less likely to be prescribed lipid-lowering agents. Yet, controlling for these conditions is difficult, If not impossible, using observational data. But, these conditions are likely to be highly related to adverse effects following a pneumonia.
For example, using a cohort of previous AMI patients, Polgreen et al. “Increased Statin Prescribing Does not Lower Pneumonia Risk” CID 2015 find similar results to the authors of the current manuscript using similar methods. However, using instrumental variables methods, they find different results. This suggests that unmeasured confounding exists. In this paper all of the patients should have been on a statin.
This health bias limitation does not diminish the hard and interesting work that the authors did and should not lower the importance of the results. However, the possibility of a “healthy user bias” needs to be stressed more in the limitations section. I do not think that the authors need to do any more analysis. For example, I can see no reason for them to do a propensity score. Instead, they merely need to expand their limitations section.
2. The authors call for randomized controlled trials to validate their results. However, a few have been done. Two find no effect of statins, (Papazian et al. Effect of statin therapy on mortality in patients with ventilator-associated pneumonia: a randomized clinical trial. JAMA 2013; Viasus et al. The effect of simvastatin on inflammatory cytokines in community-acquired pneumonia: a randomized, double-blind, placebo-controlled trial. BMJ Open 2015) and one finds very small effects, (Novack, The effect of rosuvastatin on incident pneumonia: results from the JUPITER trial. CMAJ 2012). These studies are not based in outpatient setting but should be referenced and discussed.
Minor problems:
There are some poorly constructed sentences that should be revised:
A. In the Methods section of the abstract “Multivariable logistic regression and Cox regression WERE used for analysis”
B. In the Methods section of the manuscript “…to record all instances of clinical DIAGNOSES, morbidity events…”
C. “Drug covariates adjusted for included…” should be changed to “We adjusted for the following drugs…” or something similar.
D. “…where >5% of missing data” should be changed to “… for variables with >5% of data are missing….”

·

Basic reporting

The authors do a very nice job explaining the theoretical mechanism that may exist between statin use and MAARI-related mortality. However, authors do a poor job of explaining how/why observed variation in statin use prior to MAARI provides sufficient variation to yield informative estimates of the relationships between statin use and MAARI outcomes. I believe the study results have merit only because I can imagine an identification strategy (prior statin use thought to be unrelated to outcomes associated with a future acute condition). But it is not my job to either imagine or describe the identification strategy employed. The authors need to directly state the strategy a priori (in the introduction or methods sections) along with all assumptions associated with the strategy. In the discussion section the authors then need to describe the limits of their inferences in the context of this a priori described strategy (not ex post).

Experimental design

• Studies of treatment effect such as this must directly state its objective in the introduction section along with their identification strategy to estimate parameters that properly assess this objective. Are the authors attempting to assess the average treatment effect (ATE) of prior statin use across all MAARI patients? Or the average treatment effect across the patients treated with statins (ATT)? Note that regression-based observational studies like this, even without confounding problems, only identify ATT (Heckman JJ, Robb R. Alternative methods for evaluating the impact of interventions. In: Heckman JJ, Singer B, eds. Longitudinal Analysis of Labor Market Data. New York, NY: Cambridge University Press; 1985:156–245). Additional assumptions are required to make inferences about untreated patients from these estimates (homogeneity, non-essential heterogeneity) and these need to be stated a priori.

Because statin prescribing is a choice, the authors need to include in the introduction a description of the factors that they theorize are related to statin prescribing and whether these factors are or are not related to study outcomes. Then in the Methods section the authors should state which of these factors are measured and which are unmeasured. And given those unmeasured, state a priori the assumptions required to obtain unbiased estimates of study objectives. For example, one might expect that statin users prior to MAARI, relative to non-statin users, would have greater preferences for maintaining their health and consume more healthcare and live healthier lifestyles. Could health preferences be also related to MAARI outcomes? Can health preferences be measured and controlled for in the analysis?

Validity of the findings

• Why did the authors choose to measure and control for the covariates specified? Because everyone else does? Or does theory suggest these variables are related to both statin choice and outcomes? What theory?

• Table 1 should contain a comparison of the patients treated with statins vs. those not treated with statins to support the identification strategy. Table 1 as specified provides little helpful information to support the study conclusions.

• In this NEW Table 1 that compares prior statin-users to prior non-statin users I personally would like to see measures of prior healthcare utilization especially measures related to preventive treatments and concurrent measures of treatments used to treat the MAARI. The authors appear to making the assumption (unstated in the intro) that statin-users and non-statin users have similar distributions of healthcare utilization other than statins. This can be easily assessed.

• Authors need to be more specific about the time periods over which study covariates were measured.

---

## Round 0.2 · Minor Revisions

Please incorporate the changes suggested by Reviewer 2.

Reviewer 1 ·

Basic reporting

The main concerns are adequately responded. The revised paper is acceptable in scientific view.

Experimental design

No comments.

Validity of the findings

No comments!

Additional comments

No comments!

Reviewer 2 ·

Basic reporting

No Comments

Experimental design

No Comments

Validity of the findings

The authors need to address the studies that did not find statins protective against pneumonia more fully. First. The authors have cited some of the randomized trials that this reviewer suggested, but they are too dismissive of the results of these trials. Second, the authors said that they read the instrumental variables paper on the same topic. They said that they found it interesting but did not cite it. They merely stated that they will not be able to do an instrumental variables analysis because they used a different outcome. (It is not true that there are no instruments that could be used in the current study: local statin treatment rates would be an excellent instrument.) It is not necessary to do an instrumental variables analysis, and I made this clear in my first review, but addressing the fact that there are different results in the literature is important for readers to know. The authors need to cite this paper, and in their limitations section they should note that they could not do an instrumental variable analysis.

Additional comments

No Comments

---

## Round 0.3 · accepted · Accept

The authors have addressed the concerns of Reviewer 2.